# Cu(In,Ga)Se_2_ Solar Cells Integrated with Subwavelength Structured Cover Glass Fabricated by One-Step Self-Masked Etching

**DOI:** 10.3390/mi11090877

**Published:** 2020-09-21

**Authors:** Ho-Jung Jeong, Ye-Chan Kim, Sung-Tae Kim, Min-Ho Choi, Young-Hyun Song, Ju-Hyung Yun, Min-Su Park, Jae-Hyung Jang

**Affiliations:** 1Photoconversion Material Research Center, Korea Photonics Technology Institute, Gwangju 61007, Korea; hojung@kopti.re.kr (H.-J.J.); songyh83@kopti.re.kr (Y.-H.S.); 2School of Electrical Engineering and Computer Science, Gwangju Institute of Science and Technology, Gwangju 61005, Korea; yckim@gist.ac.kr (Y.-C.K.); sungtaekim@gist.ac.kr (S.-T.K.); minho522@gist.ac.kr (M.-H.C.); 3Department of Electrical Engineering, Incheon National University, Yeonsu-gu, Incheon 22012, Korea; juhyungyun@inu.ac.kr; 4Department of Electronics Engineering, Dong-A University, Saha-gu, Busan 49315, Korea

**Keywords:** subwavelength structure, solar cells, self-masked etching, optical reflection loss, energy harvesting

## Abstract

We report an anti-reflective cover glass for Cu(In,Ga)Se_2_ (CIGS) thin film solar cells. Subwavelength structures (SWSs) were fabricated on top of a cover glass using one-step self-masked etching. The etching method resulted in dense whiskers with high aspect ratio. The produced structure exhibited excellent anti-reflective properties over a broad wavelength range, from the ultraviolet to the near infrared. Compared to a flat-surface glass, the average transmittance of the glass integrated with the SWSs improved from 92.4% to 95.2%. When the cover glass integrated with the SWSs was mounted onto the top of a CIGS device, the short-circuit current and the efficiency of the solar cell were enhanced by 4.38 and 6%, respectively, compared with a CIGS solar cell without cover glass.

## 1. Introduction

Singe- or multi-antireflection coating (ARC) layers have been widely utilized to reduce optical reflection losses at surface of solar cells. Generally, the ARC layers are designed to reduce reflection losses at visible wavelengths, which is the high spectral response region. Even though ARCs exhibit excellent anti-reflective characteristics, broadband antireflection, including in the visible and near-infrared wavelength regions, is necessary to maximize the efficiency of solar cells.

For this purpose, subwavelength structures (SWSs) and their broadband anti-reflective properties have attracted interest for photovoltaic applications [1,2,3,4,5,6,7]. When the period of the SWS is smaller than the wavelength of the incident light, the light diffraction can be suppressed. The generated structures provide gradual refractive index profiles between the air and the surface of solar cells over a broad wavelength range, leading to minimized Fresnel reflection loss.

Electron-beam lithography [8,9,10,11,12,13] and nano-imprint lithography [14,15,16] have been typically used to fabricate nanoscale SWSs. However, these fabrication methods suffer from some drawbacks including the complex process, high cost and small area fabrication. To overcome the problems, dry etching using the various metal dewetting techniques has been introduced for generating the SWSs [17,18,19,20,21,22]. The anti-reflective surface is made by dewetting a thin metal film, then use the metal particles for an etch mask to fabricate SWSs on a large area. Moreover, the different one-step self-masked etching methods have been reported [23,24,25,26]. The approach on which it is based, that some polymers and nonvolatile residues randomly deposited on the sample surface can act as a micro-etching mask during dry etching under certain conditions [27,28]. Thus, the nano-scaled SWSs can be formed by a one-step, maskless and inexpensive process. These etching techniques can produce stochastic nanostructures on various substrates without the complex lithography process. We conducted a comparative study to apply both approaches to a Cu(In,Ga)Se_2_ (CIGS) thin film solar cell for the first time.

This study demonstrated that incorporating a simple and low-cost SWS fabrication in a cover glass on a CIGS device resulted in the improved optical performances of the solar cell. For a comparative study, a one-step self-masked etching and a dry etching using Ag nanoparticles were conducted to fabricate the SWSs and their optical properties were compared and analyzed. The cover glass with the SWSs exhibited superior broadband anti-reflective properties from the ultraviolet (UV) to the near infrared wavelength range. The fabricated cover glasses were applied to the top of the CIGS solar cell and then device characteristics were investigated.

## 2. Materials and Methods 

Molybdenum was sputtered on soda lime glass as a back contact layer. A p-type CIGS absorber layer with a thickness of 2 μm was prepared using a three-stage co-evaporation method [29]. Afterwards, an n-type CdS buffer layer was grown on the CIGS layer by chemical bath deposition. A transparent conducting window layer consisting of intrinsic ZnO and Al-doped ZnO was deposited by RF sputtering. Finally, the front contact was formed using e-beam evaporation with a 1-μm-thick Al. The final cell area was 0.49 cm^2^, as defined by mechanical scribing.

Borosilicate glass was prepared as the cover glass for the CIGS solar cells. The glass substrates were sequentially cleaned by ultrasonic agitator in acetone, methanol and deionized water for 20 min. For the comparison, dry etching using thermally dewetted Ag nanoparticles and one-step self-masked etching were conducted to form SWSs on the top surface of the cover glasses. A 5-nm-thick Ag thin film was deposited on the surface of the cover glasses by e-beam evaporation at a deposition rate of 0.2 Å/s. A thermal treatment was carried out at 300 °C for 10 min under a nitrogen gas of 30 sccm, followed by reactive ion etching (RIE) with a CF_4_ gas flow rate of 100 sccm, RF power of 75 W and chamber pressure of 70 mTorr. To produce desirable anti-reflective SWSs on the cover glass, the etching time was controlled between 9 and 18 min. The glasses were dipped into HNO_3_ for 1 min to remove the residual Ag particles and solvent-cleaned again. For the one-step self-masked etching, the bare glass was etched under the same RIE condition. This borosilicate glass contains metal elements such as aluminum and sodium. The nonvolatile compounds of fluorine and the metals are created in CF_4_ plasm and they work as micro-masks [28]. Therefore, the grassy etched surface was observed. The etching time was also varied from 9 to 18 min and the etched samples were cleaned by solvents.

The fabricated SWS-integrated glasses were mounted on the top of CIGS devices with a UV curing optical adhesive. The structural and morphological properties of the SWSs were observed by field-emission scanning electron microscope (FE-SEM). The optical properties of the SWS-integrated glasses and CIGS devices were analyzed using UV–VIS–NIR spectroscopy (Cary 5000, Varian, Santa Clara, CA, USA) with an integrating sphere. The photovoltaic performance of the CIGS devices was measured using a solar simulator (Oriel Sol3A Class AAA, Newport Corporation, Irvine, CA, USA) and incident photon to electron conversion efficiency (QEX7 series, PV measurements, Inc., Point Roberts, WA, USA).

## 3. Results and Discussion

Figure 1 shows a schematic diagram and photographs of the CIGS solar cell with the cover glass. The cover glass was designed with dimensions of 5.7 × 8 × 0.7 mm^3^. Figure 2 shows SEM images of the Ag thin films on the glass substrate: (a) the bare glass, (b) the as-deposited glass with a thickness of 5 nm and (c) the deposited glass after annealing at 300 °C in N_2_ atmosphere for 10 min. During the annealing process, the Ag nanoparticles were formed due to agglomeration of the Ag thin film. The diameter distribution histogram of the Ag nanoparticles is shown in Figure 2d and average size of the particles was estimated to be 42.5 nm. Figure 3 shows SEM images of the SWS-integrated cover glasses fabricated by (a) dry etching using the Ag nanoparticles for the etch mask and (b) one-step self-masked etching with the various etching times from 9 to 18 min. The insets are corresponding 20° tilted cross-section SEM images.

The SEM images confirm that the fabricated SWSs formed in a disordered manner on the surface of the cover glass. In addition, the morphology of the etched profile was changed by the two different etching methods. The SWSs fabricated by the dry etching using the Ag nanoparticles had cone-shaped structures. On the other hand, the pillar-shaped SWSs were formed by the one-step self-masked etching. As the etching time increased, the height of both SWSs also increased. However, the period of structures fabricated by the self-masked etching was shorter than that of the dry etching using the Ag nanoparticles. The densest and highest aspect ratios of the SWSs were observed on the glass substrate fabricated with the one-step self-masked etching time of 18 min. When the etching time was increased above 18 min, there was no apparent change in the height of the structures, except for increased sidewall erosion. As the aspect ratio increased, more reactive ions hit the sidewall of the SWSs, and less ions were available to etch the bottom of the structures. By considering effective medium theory, the additionally etched sidewall led to SWSs with a less optimized graded refractive index profile [30].

The measured transmittance spectra of the SWS-integrated cover glasses in the wavelength range from 300 to 1200 nm is shown in Figure 4. The transmittance of the flat surface glass is also indicated as a reference. The optical properties of the SWS-integrated cover glasses for various etching times were analyzed. The inset tables exhibit the average transmittance values of the flat surface and the SWSs-integrated glasses. Figure 4a shows the transmittance spectra of the SWS-integrated glasses fabricated by dry etching using the Ag nanoparticles. The flat surface glass had an average transmittance of 92.4% in the wavelength region between 300 and 1200 nm. The cover glass with the SWSs exhibited a higher transmittance value than the flat surface glass. The glasses with SWSs etched for 9, 12 and 15 min showed better optical transmittances, of 94.0%, 94.2% and 94.4%, in that range. When the etching time was further increased to 18 min, however, a reduction in average transmittance was observed. This indicates that the more highly etched sidewall degrades the optical properties of the SWSs in the wavelength region from 300 to 1200 nm.

Likewise, as shown in Figure 4b, the cover glass with the SWSs prepared by the one-step self-masked etching exhibited enhanced transmittance. The average transmittance of the SWS-integrated cover glasses increased as etching time increased from 9 to 18 min. They showed broadband anti-reflective properties, and the highest average transmittance of 95.2% was achieved with a self-masked etching time of 18 min. This means that the diffraction effects in the fabricated SWSs are negligible in the wavelength range from 300 to 1200 nm. This is enough to satisfy the zero-order diffraction condition.

Even though the SWS-integrated cover glasses fabricated by the dry etching using Ag nanoparticles showed the most improved transmittance properties, a slight decrease in transmittance was observed at wavelengths over 500 nm. On the other hand, the superior optical transmittance of the SWS-integrated cover glass was measured over the entire wavelength range when it was prepared using the one-step self-masked etching. The dense and high aspect ratio of the SWSs improved the broadband anti-reflective characteristics and facilitated light absorption into the solar cells.

Figure 5 shows the reflectance spectra and the external quantum efficiency (EQE) of CIGS devices with the flat surface and the SWS-integrated cover glass with the one-step self-masked etching time of 18 min, respectively. The reflectance spectra optical losses were reduced by introducing the SWSs onto the surface of the cover glass. Compared with the CIGS device without the cover glass, the average reflectance of the CIGS device with SWS-integrated cover glass was reduced from 14.3 to 8.0% in the wavelength range between 300 and 1200 nm. The measured EQE shows increased quantum efficiency after employing the SWSs-integrated cover glass. The enhancement in the EQE can be ascribed to reduced surface reflection and increased photon harvesting.

To estimate the influence of the reflectance of the SWSs on the performance of the CIGS solar cells in relation to the incident solar spectrum, the solar weighted reflectance (SWR) was calculated using the following equation [31]:(1)$$\mathrm{SWR}=\frac{\int^{\;}F\left(\lambda \right)R\left(\lambda \right)IQE\left(\lambda \right)d\lambda }{\int^{\;}F\left(\lambda \right)IQE\left(\lambda \right)d\lambda }$$
where *F*(λ) is the photon flux in the air, mass 1.5 global (AM 1.5G) spectrum, *R*(λ) is the surface reflectance and *IQE*(λ) is the internal quantum efficiency of each device. IQE(λ) is determined by the relative percentage of photo-generated electron–hole pairs lost to recombination after accounting for reflection at the surface and can be expressed as
(2)$$IQE\left(\lambda \right)=\frac{EQE\left(\lambda \right)}{1-R\left(\lambda \right)}$$

As expected, the calculated SWRs of the CIGS devices with the flat surface and the SWSs-integrated cover glass were decreased by 3.5% and 6.2%, respectively, compared with the reference device. These results demonstrate that employing the SWSs-integrated cover glass on the CIGS solar device facilitates light absorption for generating photocurrent.

The current density–voltage characteristics (J–V) of the CIGS devices with the flat and the SWS-integrated cover glass under AM 1.5G illumination are shown in Figure 6. The inset tables exhibit the photovoltaic parameters of the solar cells. Each device is compared with the reference cell without a cover glass. The short-circuit current density (*J*_sc_) and efficiency (*η*) of the CIGS cell with the flat surface glass were slightly improved, by 3.4 ± 0.05 and 2.6 ± 0.04%, respectively, compared with those of the reference cell. On the other hand, the CIGS cell with the SWS-integrated cover glass exhibited relative enhancement in *J*_sc_ and *η* of 4.3 ± 0.08 and 6±0.06%, respectively, for five different CIGS cells. By introducing the SWS-integrated cover glass on the CIGS solar cell, the generated current and efficiency was increased due to the reduction in surface reflection. The open-circuit voltage (*V*_oc_) and fill factor (FF) remained essentially the same.

The above results confirm that the SWSs fabricated using one-step self-masked etching can minimize the optical losses taking place on the surface of the cover glass, finally leading to the enhanced performance of the CIGS solar cell.

## 4. Conclusions

SWSs were applied to a cover glass using a simple, fast and cost-effective fabrication method. Compared with the technique of Ag nanoparticle-based dry etching, SWSs fabricated using a one-step self-masked etching had a higher aspect ratio and superior broadband anti-reflective properties. When the SWS-integrated cover glass was implemented on the top of a CIGS device, the performance of the solar cell was highly improved. The increase in *J*_sc_ and *η* was caused by the minimization of reflection losses at the surface of the cover glass.

## Figures and Tables

**Figure 1 micromachines-11-00877-f001:**
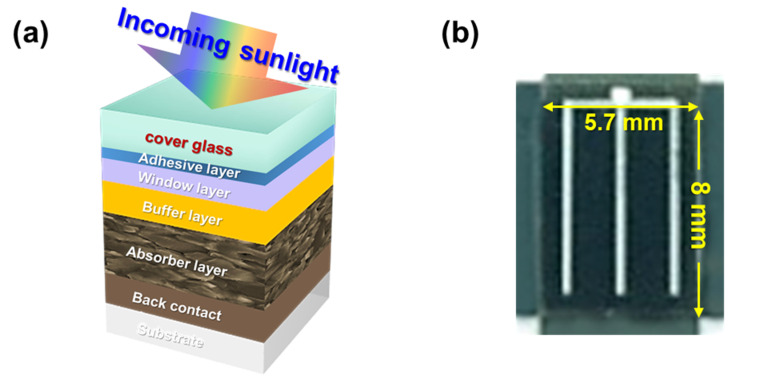
(**a**) Schematic diagram and (**b**) photograph of a Cu(In,Ga)Se2 (CIGS) thin film solar cell integrated with a cover glass.

**Figure 2 micromachines-11-00877-f002:**
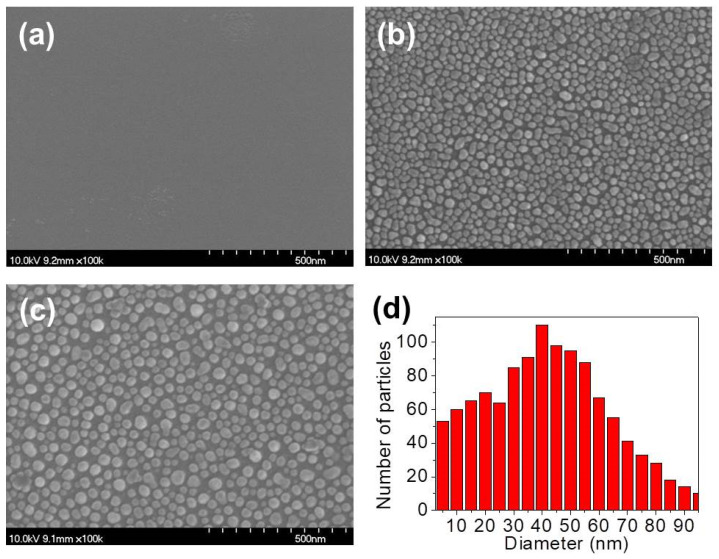
Surface SEM images of (**a**) the bare glass, (**b**) the 5-nm-thick Ag thin film and (**c**) the Ag nanoparticles after annealing at 300 °C for 10 min. (**d**) Diameter distribution of the Ag nanoparticles.

**Figure 3 micromachines-11-00877-f003:**
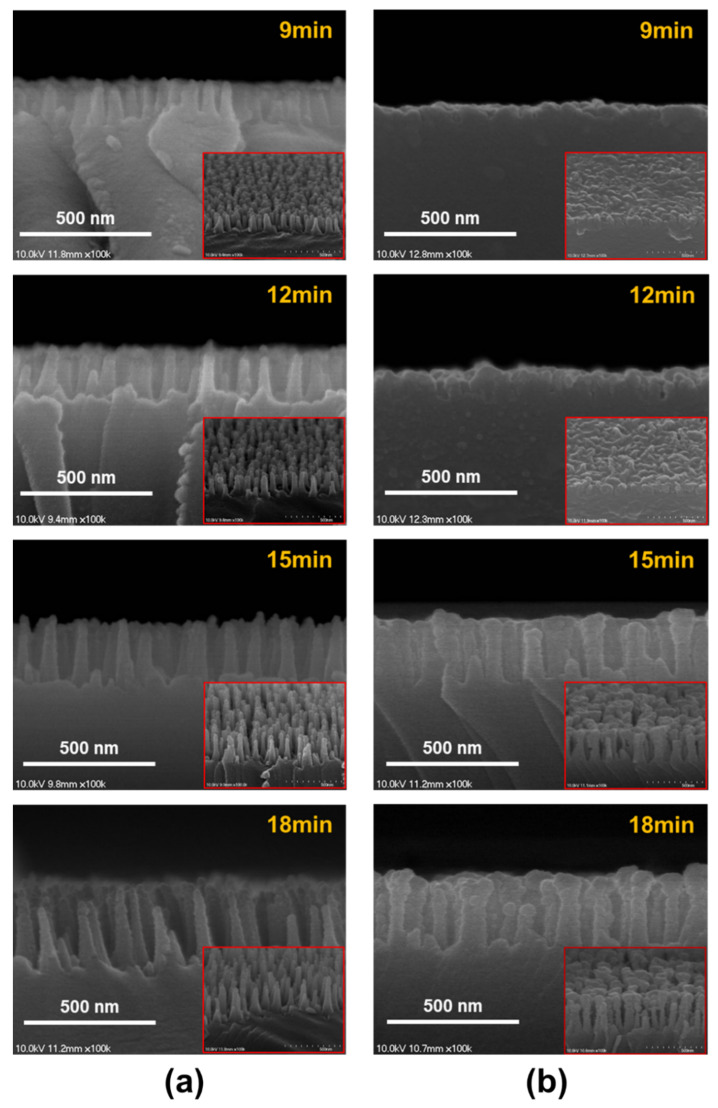
SEM images of the subwavelength structure (SWS)-integrated cover glass fabricated by (**a**) dry etching using the Ag nanoparticles and (**b**) one-step self-masked etching with various etching times.

**Figure 4 micromachines-11-00877-f004:**
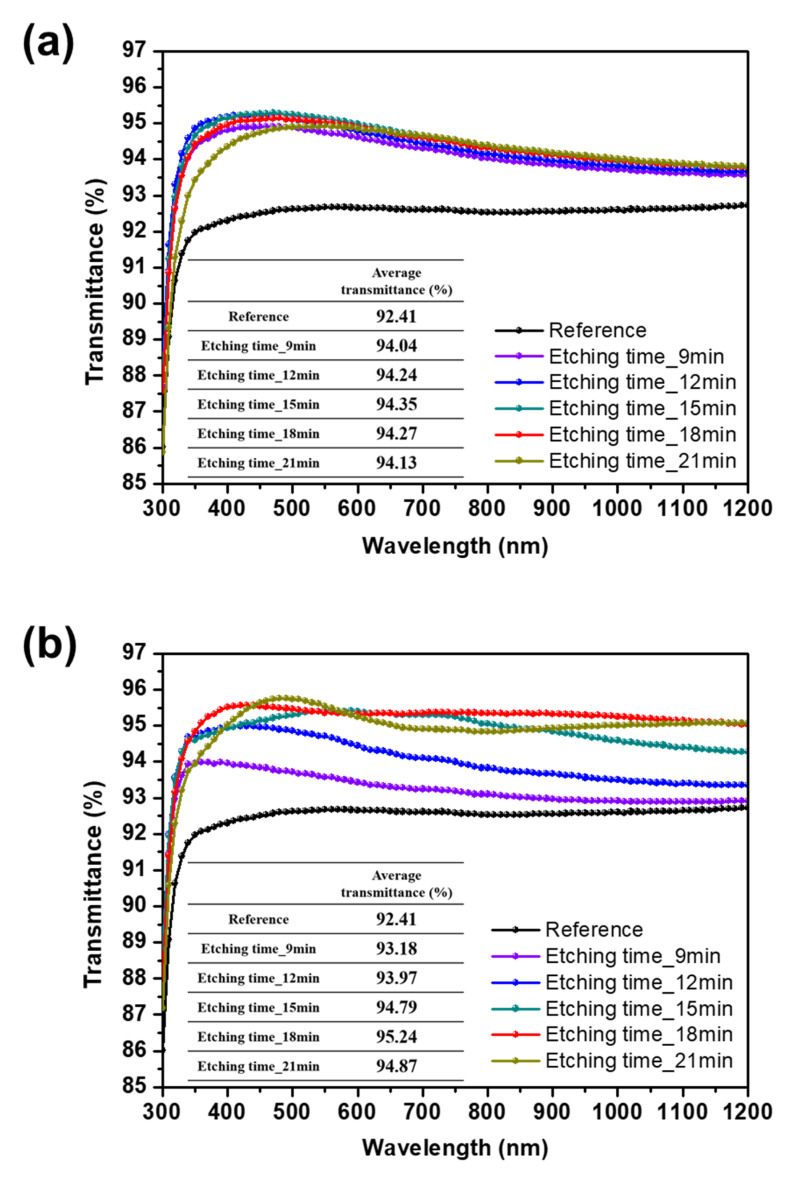
The measured transmittance spectra of the flat surface cover glass (reference) and SWS-integrated cover glass fabricated by (**a**) dry etching using the Ag nanoparticles and (**b**) the one-step self-masked etching with the etching time of 9–18 min.

**Figure 5 micromachines-11-00877-f005:**
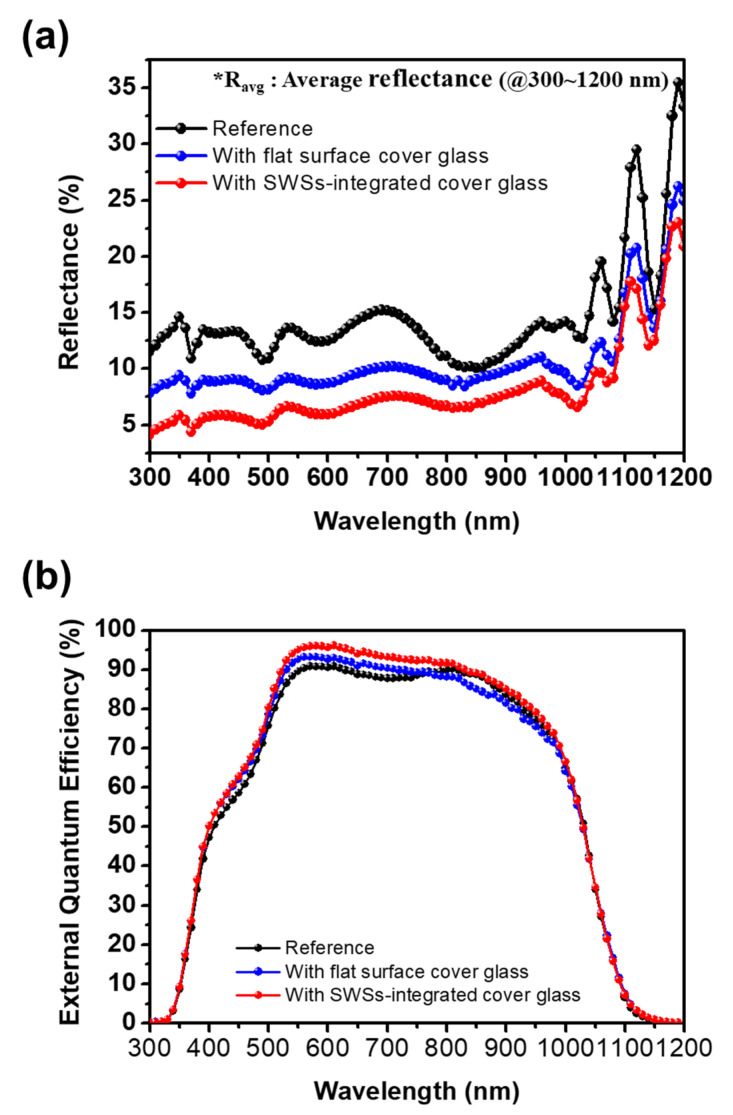
The measured (**a**) reflectance and (**b**) external quantum efficiency (EQE) spectra of the CIGS devices with the flat surface and the SWS-integrated cover glass, as compared with those of the reference device (without cover glass).

**Figure 6 micromachines-11-00877-f006:**
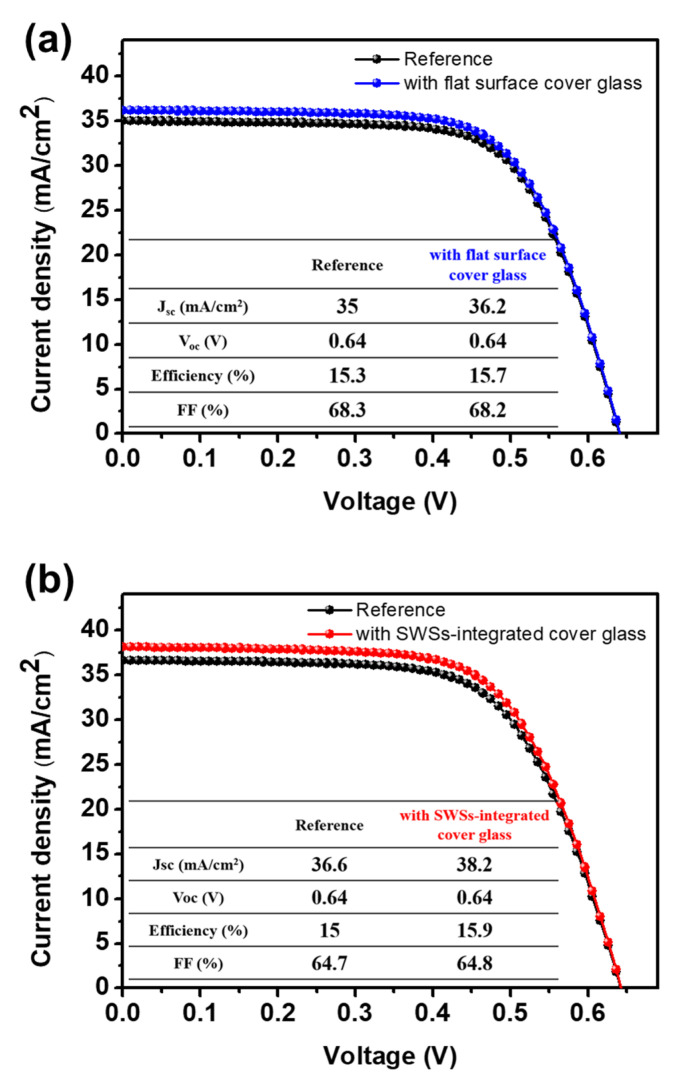
Current density–voltage curves of the CIGS devices with (**a**) the flat surface glass and (**b**) the SWS-integrated cover glass, as compared with those of the reference devices.

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
