# Peer review of "Cu(In,Ga)Se2 Solar Cells Integrated with Subwavelength Structured Cover Glass Fabricated by One-Step Self-Masked Etching"

_micromachines, 2020, doi:10.3390/mi11090877_

Round 1
Reviewer 1 Report
- Authors should clearly describe the conditions for one-step self-masked etching. If the conditions are the same as for dry etching using Ag nanoparticles, then the authors should write how self-masked etching occurs, since no polymer is formed from the plasma of pure CF4 [J. W. Coburn and Harold F. Winters. Plasma etching—A discussion of mechanisms. Journal of Vacuum Science & Technology 16, 391 (1979); doi: 10.1116/1.569958. Pages 397-398.].
- There is a typo on line 97: "nanaoparticles" should be replaced with "nanoparticles".
Reviewer 2 Report
In this manuscript, the authors proposed a facile approach to synthesize subwavelength structures for the fabrication of CIGS solar cells, and the efficiency of the solar cells integrated with the SWSs was enhanced by 6% owing to the improved average transmittance from 92.4% to 95.2%. The results are well presented and analyzed in a systematic manner with good English. Thus, the referee considers that the submitted results will give readers a valuable aid point and further opportunity for the investigations in this field. Therefore, the experimental work and the discussion presented by the authors are of fine quality & scope in order to be suitable for the publication in this journal, which is required a major revision before publication as shown below.
Comment 1:The author failed to clarify the difference between one-step self-masked etching and direct dry etching. Did difference come from the annealing. As reported by many works, the configuration of metallic nanoparticles can be controlled by annealing temperature (i.e. DOI: 10.1039/c5ce02439k, DOI 10.1186/s11671-015-1084-z). It’s better to show the annealing temperature effect on the etching process.
Comment 2:In Figure3, there are no labels indicate (a) and (b).
Comment 3:The average transmittance was slightly improved by only 2.8% from 92.4% to 95.2%, which was not too noticeable within error range. The author is suggested to provide the reproducibility of the transmittance and efficiency of the resulting solar cells.
Comment 4:It’s better to provide full term of CIGS solar cells to improve the clarity, although it has been widely known.
Comment 5:The author should give a reasonable explanation of the morphological difference after etching via for two etching method, as the self-mask was derived from Ag nanoparticles after annealing.
Comment 6:There are many types and errors appeared in current version. Therefor, a significant modification is required for the revised manuscript.
Round 2
Reviewer 2 Report
The manuscript can be accepted in current shape